# EFFICIENT OFFLINE REINFORCEMENT LEARNING: THE CRITIC IS CRITICAL

## ABSTRACT

Recent work has demonstrated both benefits and limitations from using supervised approaches (without temporal-difference learning) for offline reinforcement learning. While off-policy reinforcement learning provides a promising approach for improving performance beyond supervised approaches, we observe that training is often inefficient and unstable due to temporal difference bootstrapping. In this paper we propose a best-of-both approach by first learning the behaviour policy and critic with supervised learning, before improving with off-policy reinforcement learning. Crucially, we demonstrate that the critic can be learned by pre-training with a supervised Monte-Carlo value-error, making use of commonly neglected downstream information from the provided offline trajectories. This provides consistent initial values for efficient improvement with temporal difference learning. We further generalise our approach to entropy-regularised reinforcement learning and apply our proposed pre-training to state-of-the-art hard and soft off-policy algorithms. We find that we are able to more than halve the training time of the considered offline algorithms on standard benchmarks, and surprisingly also achieve greater stability. We further build on our insight into the importance of having consistent policy and value functions to propose novel hybrid algorithms that regularise *both* the actor and the critic towards the behaviour policy. This maintains the benefits of pre-training when learning from limited human demonstrations.

## 1 INTRODUCTION

Recent work has highlighted the effectiveness of supervised learning approaches (without temporal difference learning) for offline reinforcement learning (Emmons et al., 2022; Chen et al., 2021; Brandfonbrener et al., 2021; Peng et al., 2019). Other work has analysed in detail the limitations of these supervised approaches and when off-policy reinforcement learning techniques should be favoured (Kumar et al., 2022b; Brandfonbrener et al., 2022; Paster et al., 2022). Given these seemingly opposing approaches, it is natural to ask: can we get the best of both supervised learning and temporal difference learning for offline reinforcement learning? Specifically, can we provide an approach that provides the training efficiency and stability of supervised learning, while still gaining the performance benefits of multi-step temporal difference learning? In this work we investigate such an approach, by supervised pre-training off-policy reinforcement algorithms to be consistent with the behaviour policy and values before attempting improvement. We utilise the entire offline dataset to do so, and find significant efficiency and stability benefits. While pre-training a policy with behaviour cloning has been considered previously (Zhang & Ma, 2018; Hill et al., 2018; Pomerleau, 1991), our core contribution is to demonstrate that by appropriately pre-training the critic with a Monte-Carlo value-error objective to get *consistent* actor and critic networks for the behaviour data, the initial consistency of the critic evaluations provides more stable temporal difference updates that lead to significantly improved training efficiency and stability, even when taking the cost of pre-training into account.

As a motivational example, we can consider offline tabular $Q$-learning on the simple 4 state MDP illustrated below, where we are provided with a single offline trajectory from an unknown policy.

Tabular $Q$-learning initialises all $Q$-values to zero (or equivalently $Q$-networks are randomly initialized such that all initial $Q$-values are close to zero for deep $Q$-learning), and then performs temporal difference updates using the following update rule (Sutton et al., 2018):

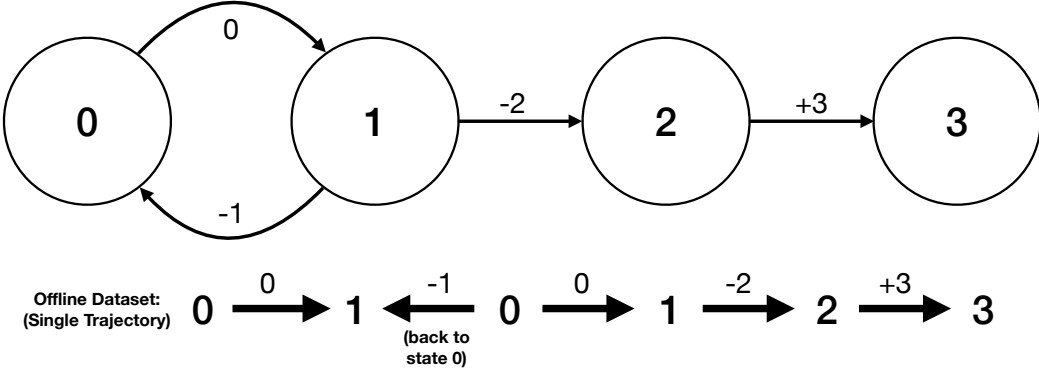

Figure 1: A motivational tabular MDP. In offline reinforcement learning, we are provided with a dataset of trajectories. In this paper we utilise information from the entire trajectory to initialise a self-consistent critic for off-policy reinforcement learning, which eliminates much of the initial inefficiency and instability associated with bootstrapping in temporal difference losses.

$$Q(s_t, a_t) \leftarrow Q(s_t, a_t) + \alpha \left[ r_t + \gamma \max_a Q(s_{t+1}, a) - Q(s_t, a_t) \right] \tag{1}$$

For simplicity in this minimal motivational example, let us take the discount factor $\gamma$ and the learning rate $\alpha$ to be 1. Performing updates using the provided offline trajectory leads to the $Q$-value updates shown in the left column of Table 1. We find that the policy converges to the optimal policy after 2 epochs, and the values converge to the correct optimal values after 3 epochs.

However, in the case of offline reinforcement learning, we have access to better initial $Q$-value estimates in the form of Monte-Carlo samples from the trajectory. This follows since in addition to Bellman's optimally equation in Equation 1, we also have the definition of the Q-function for the behaviour policy $\pi_B$ for which we can compute a sample-based expectation from our offline data:

$$Q^{\pi_B}(s_t, a_t) = \mathbb{E}_{\tau \sim \pi_B} \left[ \sum_{n=0}^{\infty} \gamma^n r_{t+n} \Big| s_t, a_t \right] \tag{2}$$

Initialising the $Q$-values with Monte-Carlo estimates sampled from the provided trajectory and using the same update rule given in Equation 1 leads to the $Q$-values shown in the right column of Table 1. We now find that the policy converges to the optimal policy immediately after initialisation, and the values converge to the optimal values after a single epoch. This improved efficiency is a result of initialising the temporal difference updates with values that incorporate additional information beyond a single step, which would usually take multiple passes through the data to propagate.

Table 1: State-action values for each epoch of Q-learning to convergence for the motivational MDP and offline trajectory provided in Figure 1 for both zero and Monte-Carlo value initialisations.

| Epoch | Zero Initialisation | | | | MC Value Initialisation | | | |
|---|---|---|---|---|---|---|---|---|
| | $Q(0, \rightarrow)$ | $Q(1, \leftarrow)$ | $Q(1, \rightarrow)$ | $Q(2, \rightarrow)$ | $Q(0, \rightarrow)$ | $Q(1, \leftarrow)$ | $Q(1, \rightarrow)$ | $Q(2, \rightarrow)$ |
| 0 | 0 | 0 | 0 | 0 | 0 or +0.5[1] | 0 | +1 | +3 |
| 1 | 0 | -1 | -2 | +3 | +1 | 0 | +1 | +3 |
| 2 | -2 | -2 | +1 | +3 | +1 | 0 | +1 | +3 |
| 3 | +1 | 0 | +1 | +3 | +1 | 0 | +1 | +3 |

This minimal example demonstrates that the benefit of pre-training values on offline data arises even in the absence of function approximation, near-optimal behaviour policies, or complex MDPs. However, the inefficiency of boostrapping from randomly initialized temporal difference (TD) targets still exists in the case of uninformed neural network initializations, and in sparse-reward environments where rewards received at the end of long trajectories may take many TD updates to propagate.

---

[1]The MC initialisation value for $Q(0, \rightarrow)$ depends on the choice of first-visit or every-visit Monte-Carlo (Sutton et al., 2018). In this example it doesn't matter which is used after initialisation.

In this paper, we provide similar approaches for pre-training a critic or value network, which can be combined with most existing off-policy reinforcement learning algorithms, leading to improved training efficiency and stability. We first consider return-maximising reinforcement learning, and apply our approach to a minimal offline reinforcement learning baseline, TD3+BC (Fujimoto & Gu, 2021). We then generalize our procedure for entropy-regularised reinforcement learning, and apply our approach to a state-of-the-art offline baseline, EDAC (An et al., 2021). Finally, we build on our insight into the importance of having consistent policy and value functions by introducing novel hybrid algorithms, TD3+BC+CQL and EDAC+BC, that regularise *both* the actor and the critic towards the behaviour policy to maintain the benefits of pre-training and demonstrate performance improvements on the challenging Adroit environments (Rajeswaran et al., 2018).

## 2 PRELIMINARIES AND RELATED WORK

### 2.1 PRELIMINARIES

Online reinforcement learning (RL) involves an agent taking actions according to a policy $\pi$ to interact with a Markov Decision Process (MDP). An MDP can be defined by the tuple $(\mathcal{S}, \mathcal{A}, \mathcal{T}, r, d_0, \gamma)$ where $\mathcal{S}$ is the state space, $\mathcal{A}$ is the action space, $\mathcal{T}(s'|s, a)$ is the transition probability distribution, $r : \mathcal{S} \times \mathcal{A} \to \mathbb{R}$ is the reward function, $d_0$ is the distribution of initial states, and $\gamma \in (0, 1]$ is a discount factor. The goal is generally to learn the policy $\pi^*$ that maximises the expected discounted returns: $\pi^* = \arg\max_\pi \mathbb{E}_{\pi, \mathcal{T}} \left[ \sum_{t=0}^{\infty} \gamma^t r(s_t, a_t) \right]$. Offline RL poses the same goal, but the policy $\pi$ must be learned from a fixed dataset of interactions from a behaviour policy $\pi^B$, without any additional data collection. This behaviour policy $\pi^B$ is generally unknown and arbitrarily optimal, and may be a human policy, a rule-based or hardcoded policy, or another learned agent policy.

### 2.2 OFFLINE REINFORCEMENT LEARNING APPROACHES

Perhaps the most straightforward approach for learning a policy from the offline data is behaviour cloning, which utilises $(s, a)$ pairs from the dataset as input-target pairs with a supervised loss function (Pomerleau, 1991). Since the training is supervised, convergence is relatively stable and efficient, but the learned policy can at best match the performance of the behaviour policy since behaviour cloning does not utilise reward information, and online performance may be brittle due to accumulating errors taking the agent out-of-distribution (OOD) of known states (Ross et al., 2011).

What if we want to improve on the behaviour policy? Behaviour cloning variants such as BC-$k\%$ (Levine et al., 2020) and Advantage-Weighted Regression (Peng et al., 2019; Peters & Schaal, 2007) utilise reward information to selectively clone the behaviour policy. More recently, conditioning the policy on desired returns or goals (Srivastava et al., 2021; Ma et al., 2022) has seen some success with transformer based approaches Chen et al. (2021); Janner et al. (2021); Carroll et al. (2022). While these behaviour cloning variants are sometimes able to achieve generalisation to greater returns at test-time than observed in the dataset through mechanisms such as trajectory stitching (Brandfonbrener et al., 2022; Kumar et al., 2022b), in general their performance is limited by the dataset due to the absence of TD losses and extended temporal information (Paster et al., 2022; Yang et al., 2022).

One of the most promising approaches for improving on the behaviour policy is to use off-policy reinforcement learning, usually in the form of actor-critic algorithms for continuous actions (Lillicrap et al., 2019; Silver et al., 2014). For application to offline RL, the provided dataset can be loaded into the replay buffer (Mnih et al., 2015) at initialisation (Levine et al., 2020). However, naively taking the offline dataset as the replay buffer for an off-policy algorithm usually leads to policy collapse (Fujimoto et al., 2019). This occurs because as the policy (actor) and values (critic) optimise beyond the behaviour performance, they necessarily go out-of-distribution of the data (Chen & Jiang, 2019). Since there are no additional interactions to provide correcting feedback on these actions and values as there would be in the online case, growing extrapolation errors cause erroneous values and actions that lead to performance equivalent to random. Most modifications of off-policy reinforcement learning algorithms for the offline setting involve regularisation of either the actions *or* the values towards the provided dataset to prevent this out-of-distribution extrapolation (Fu et al., 2022). TD3+BC (Fujimoto & Gu, 2021) modifies TD3 (Fujimoto et al., 2018) by introducing a behaviour cloning term to regularise the policy towards the behaviour policy. Alternatively, CQL (Kumar et al., 2020) modifies Q-learning to regularise the values for out-of-distribution actions to

prevent positive extrapolation error. However, since regularisation towards the behaviour policy or values limits performance improvement (Moskovitz et al., 2022), recent approaches instead aim to capture out-of-distribution uncertainty (Wu et al., 2021). SAC-N and EDAC (An et al., 2021) use the minimum of an ensemble of critics to obtain value estimates that minimise positive extrapolation error (with EDAC introducing additional diversification loss over SAC-N to reduce the required ensemble size), such that policy optimisation is less likely to lead to policy collapse.

## 2.3 USE OF MONTE-CARLO VALUES AND PRE-TRAINING IN REINFORCEMENT LEARNING

While these off-policy RL approaches can lead to better performance than modified behaviour cloning approaches, their convergence can be inefficient and unstable due to the bootstrapping in the Bellman equation used for learning value functions. Recent work (Fujimoto et al., 2022; Patterson et al., 2022; Chen & Jiang, 2019) has demonstrated that the Bellman error can be a poor proxy for the real value error, particularly when used for incomplete, off-policy datasets as in the offline setting, causing significant issues with utilising the Bellman error as the objective for training value functions offline. Monte-Carlo (MC) return estimates have previously been used successfully in online reinforcement learning to improve the sample efficiency of online exploration (Bellemare et al., 2016; He et al., 2016; Ostrovski et al., 2017; Oh et al., 2018; Tang, 2021; Wilcox et al., 2022), but none of these approaches consider how to use MC returns in offline reinforcement learning, which is becoming of increasing importance for scaling reinforcement learning (Kumar et al., 2022a).

In the context of offline RL, Brandfonbrener et al. (2021) similarly recognise the effectiveness of learning the behaviour policy and value function before improvement, but do not consider the use of supervised learning to improve efficiency, and only take one step of TD improvement to prevent out-of-distribution extrapolation of this value function, rather than the more general and controllable combination of actor and critic regularisation we propose. Pre-training policies with imitation learning for offline RL was recently investigated by Orsini et al. (2021), but they found that the gain from pre-training is generally insignificant due to policy updates from randomly initialised critic networks causing the policy to rapidly deteriorate (as we find in Section 4), motivating our work on pre-training the critic. Other work has considered pre-training off-policy algorithms using expert demonstrations (Goecks et al., 2020; Zhang & Ma, 2018), but only consider the online setting and do not consider efficiency. Our work provides the first analysis of the benefits of pre-training with a supervised *value-error* objective, leading to more efficient and stable subsequent off-policy reinforcement learning.

# 3 PRETRAINING OFF-POLICY REINFORCEMENT LEARNING ALGORITHMS

## 3.1 OUTLINE PROCEDURE

We now explain our pre-training procedure to improve the computational efficiency of off-policy reinforcement learning algorithms, and begin by considering the standard return maximising case. In this setting we first compute the discounted return-to-go $R$ from each state-action pair until the end of the trajectory, and use this to augment each interaction tuple to incorporate information on future return. We then pre-train the actor with behaviour cloning and the critic or $Q$-network with the pre-computed discounted return-to-go, both using supervised mean-squared error minimisation (or cross-entropy for discrete actions). While the implicit Gaussianity assumption is sufficient for most current benchmarks, we note that for more complex datasets the behaviour policy and returns may be asymmetric or multi-modal, in which case other pre-training objectives may be required. This procedure provides consistent critic values corresponding to the behaviour policy learned by the actor. Finally, a suitable off-policy reinforcement learning algorithm (utilising a temporal difference loss) can be applied to these consistent pre-trained actor and critic networks to efficiently increase the policy return. Our pre-training procedure is outlined in pseudocode in Algorithm 1. As we will see in Section 4, this increase in training efficiency more than makes up for the time and computational expense associated with the pre-training.

## 3.2 BIAS-VARIANCE AND OPTIMISM-PESSIMISM TRADEOFFS

Under the behaviour policy, this discounted return to go provides a Monte-Carlo (MC) sample of the expectation that the critic or $Q$-network aims to predict. In the case of deterministic environments

and a single deterministic behaviour policy, this Monte Carlo sample will equal the expectation exactly. For stochastic behaviour policies and environments, this Monte Carlo sample may become high variance. In this case it is possible to use an $n$-step or $\lambda$-return to reduce this variance at the cost of bias introduced by bootstrapping (a well known case of the bias-variance tradeoff) (Sutton et al., 2018). However, since computing the $\lambda$-return would require inferring the current critic value for every downstream state in the trajectory, a more practical way of controlling this tradeoff is simply to compute the $TD(0)$ return (bootstrapping off the critic value for the next state) and combining the MC return with the TD return weighted by a tradeoff parameter $\lambda \in [0, 1]$:

$$\tilde{R} = (1 - \lambda)R + \lambda(r + \gamma Q'(s', a')) \tag{3}$$

For large offline datasets where the greatest training efficiency gains are possible, there should be sufficient Monte Carlo samples alongside the smoothing inductive bias of modern network architectures to reduce this variance to a manageable level for pre-training, so $\lambda$ can be set to $\sim 0$ in order to utilise maximal information from the offline data (i.e. the full return to go). However, for smaller datasets capturing stochastic policies and environments, larger $\lambda$ may be beneficial. We investigate the empirical effect of varying $\lambda$ in our initial experiments in Appendix A.4.

Additionally, by pre-training on sampled returns with a symmetric error, the critic is equally likely to under- or over-estimate the values of out-of-distribution actions when optimising the policy after pre-training, even in the deterministic case where the returns are exact. This overestimation can lead to policy collapse as discussed in Section 2. Therefore it can be helpful to add some value regularisation $\mathcal{R}(Q(s, a))$ during pre-training, such as that introduced in CQL (Kumar et al., 2020) to over-estimate the gap between in-distribution and out-of-distribution actions and effectively lower-bound the $Q$ function. We will see in Section 5 that including some value regularisation can be beneficial when the offline data is limited.

### 3.3 GENERALISATION TO MAXIMUM ENTROPY OFF-POLICY RL ALGORITHMS

The maximum entropy RL framework, and in particular Soft Actor-Critic (Haarnoja et al., 2019) along with offline variants such as SAC-N and EDAC (An et al., 2021), have recently become popular for their improved robustness, generalisation and sample efficiency relative to the 'hard' return maximisation considered above. This 'soft' RL framework involves maximising the expected return alongside the entropy of the policy, balanced by a temperature parameter (Ziebart et al., 2010):

$$\pi^* = \arg\max_{\pi} \mathbb{E}_{\tau \sim \pi} \left[ \sum_{t=0}^{\infty} \gamma^t r_t + \alpha \mathcal{H}(\pi(\cdot|s_t)) \right] \tag{4}$$

where $\mathcal{H}(\pi(\cdot|s_t)) = \mathbb{E}_{\tilde{a} \sim \pi(\cdot|s_t)} \left[ -\log(\pi(\tilde{a}|s)) \right]$ is the entropy of the policy $\pi$ in state $s_t$.

This also modifies the definition of the state-action value functions as follows:

$$Q(s_t, a_t) = \mathbb{E}_{\tau \sim \pi} \left[ \sum_{n=0}^{\infty} \gamma^n r_{t+n} + \alpha \sum_{n=1}^{\infty} \gamma^n \mathcal{H}(\pi(\cdot|s_{t+n})) \Big| s_t, a_t \right] \tag{5}$$

Therefore, in order to pre-train value functions for the behaviour policy as before we must modify the return-to-go to incorporate these entropy bonuses from every future timestep except the first. However, since these entropy bonuses depend on the current policy, we now separate our pre-training procedure above into two phases. First, we pretrain our policy with soft behaviour cloning:

$$\mathcal{L}_{\pi_{\theta}} = \mathbb{E}_{s, a \sim D, \tilde{a} \sim \pi} \left[ \alpha \log(\pi_{\theta}(\tilde{a}|s)) - \log(\pi_{\theta}(a|s)) \right] \tag{6}$$

This provides an approximate behaviour policy with which we can compute Monte Carlo samples of Equation 5 as soft returns-to-go, which can be used to augment the offline dataset as in Section 3.1. These soft returns-to-go can then be used as the targets to pre-train the critic to achieve consistency with the soft behaviour cloned policy, and providing a springboard initialization for a soft off-policy RL algorithm to efficiently improve the policy. The full pseudocode for this procedure is outlined in Algorithm 2, and a further discussion of the rational for this procedure is included in Appendix A.1.

---

**Algorithm 1** Pre-training Hard Off-Policy RL

**Require:** Dataset $D$ for use as replay buffer
   Initialise $\pi_\theta$ and $Q_\phi$ parameters, $\theta$ and $\phi$
   **for** each transition $(s_t, a_t, s_{t+1}, r_t) \in D$ **do**
      Compute $R_t = \sum_{n=0}^{T-t} \gamma^n r_{t+n}$
              $\triangleright$ Or $n$-step/$\lambda$-return/$\tilde{R}$ (eq. 3)
      Append $R_t$ to transition:
               $(s_t, a_t, s_{t+1}, r_t, R_t)$
   **end for**
   **while** not converged **do**     $\triangleright$ **Pre-Training**
      Sample batch $B = (s, a, s', r, R) \sim D$
      Update $\theta$ with behaviour cloning:
      $\mathcal{L}_\theta = \mathbb{E}_B \left[ (\pi(s) - a)^2 \right]$
      Update $\phi$ with (generalised) return:
      $\mathcal{L}_\phi = \mathbb{E}_B \left[ (Q(s, a) - R)^2 \right]$
                 $(+\mathcal{R}(Q(s, a))$
        $\triangleright$ Optional regularisation $\mathcal{R}(Q(s, a))$
      Polyak update target networks:
      $\phi' \leftarrow \tau\phi' + (1 - \tau)\phi$
   **end while**
   **while** $t < T$ **do**        $\triangleright$ **Off-Policy RL**
      Sample batch $(s, a, s', r, R) \sim D$
      Apply hard offline RL update to pre-trained $\pi$ and $Q$ to improve returns
   **end while**

**Algorithm 2** Pre-training Soft Off-Policy RL

**Require:** Dataset $D$ for use as replay buffer
   Initialise $\pi_\theta$ and $Q_\phi$ parameters, $\theta$ and $\phi$
   **while** not converged **do**     $\triangleright$ **Actor Pre-Training**
      Sample batch $B = (s, a, s', r, R) \sim D$
      Update $\theta$ with soft behaviour cloning:
      $\mathcal{L}_\theta = \mathbb{E}_{B, \tilde{a} \sim \pi} \left[ \alpha \log(\pi(\tilde{a}|s)) - \log(\pi(a|s)) \right]$
   **end while**
   **for** transition $(s, a, s', r) \in D$ **do**
      $R_t = \sum_{n=0}^{T-t} \gamma^n r_{t+n} + \alpha \sum_{n=1}^{T-t} \gamma^n \mathcal{H}(\pi(\cdot|s_{t+n}))$
              $\triangleright$ Or $n$-step/$\lambda$-return/$\tilde{R}$ (eq. 3)
      Append $R$ to transition: $(s, a, s', r, R)$
   **end for**
   **while** not converged **do**     $\triangleright$ **Critic Pre-Training**
      Sample batch $B = (s, a, s', r, R) \sim D$
      Update $\phi$ with (generalised) soft return:
      $\mathcal{L}_\phi = \mathbb{E}_B \left[ (Q(s, a) - R)^2 \right] (+\mathcal{R}(Q(s, a))$
             $\triangleright$ Optional value regularisation $\mathcal{R}$
      Update target networks: $\phi' \leftarrow \tau\phi' + (1 - \tau)\phi$
   **end while**
   **while** $t < T$ **do**        $\triangleright$ **Off-Policy RL**
      Sample batch $(s, a, s', r, R) \sim D$
      Apply soft RL updates to pre-trained $\pi$ and $Q$
   **end while**

---

# 4 INITIAL EXPERIMENTS ON D4RL MUJOCO

## 4.1 IMPLEMENTATION DETAILS

We begin our investigation into the benefits of pre-training off-policy RL algorithms by considering the D4RL MuJoCo benchmark (Fu et al., 2021). We utilise the standard HalfCheetah, Hopper and Walker2d environments and the medium dataset (a suboptimal policy with approximately $1/3$ of the performance of the expert) of 1M transitions, since this provides a meaningful behaviour policy to learn from at initialisation, with room for improvement with off-policy reinforcement learning. We also consider the medium-replay datasets with an identical procedure in Appendix A.5.

For our implementations we utilise the Clean Offline Reinforcement Learning codebase (CORL) (Tarasov et al., 2022), an open-source library that provides high quality implementations of deep offline reinforcement learning algorithms that have been benchmarked to match published performance measures. During our investigation into improving the efficiency of off-policy reinforcement learning algorithms, we found that introducing LayerNorm (Ba et al., 2016) into both the actor and critic networks significantly improved training efficiency and stability, independently verifying the findings of Ball et al. (2023), as demonstrated in Figure 6. Therefore for all results demonstrated in this paper, we use the benchmarked implementations and default hyperparameters found in the CORL codebase, with the new addition of a LayerNorm after every linear layer except the final one for each network. We investigate the effect of LayerNorm in more detail in Appendix A.2.

As an initial base off-policy algorithm, we select TD3+BC, since this algorithm involves pure return maximisation, and requires minimal changes to the online algorithm (only adjusting a single line of code to add behaviour cloning regularisation to the actor) (Fujimoto & Gu, 2021). We also investigate the effect of pre-training on an entropy-regularised reinforcement learning algorithm, EDAC (An et al., 2021). For this case, we include the auxiliary ensemble diversification loss as value regularisation during pre-training, to prevent the collapse of the ensemble. The online performance as a function of the number of offline updates for both TD3+BC and EDAC on the standard MuJoCo environments is demonstrated below in Figure 6.

## 4.2 RESULTS AND ANALAYSIS

Figure 2: Supervised pre-training before offline reinforcement learning is more efficient than offline reinforcement learning from scratch. Surprisingly, performance is also more stable long after the initial pre-training. Plots show mean and standard deviation at each timestep for 3 independent runs.

We find that pre-training the actor and critic as described in Algorithms 1, 2 leads to much more efficient training, both for TD3+BC (a hard RL algorithm with actor regularisation) and for EDAC (a soft RL algorithm with critic regularisation), even when taking the cost of pre-training into account. In particular, we find that on the more complex Hopper and Walker2d environments, **the inclusion of pre-training generally reaches expert-level performance in less than** $1/2$ **of the training steps and computation time required without pre-training** (and in less than $1/5$ of the training steps and computation time of the currently used implementations without LayerNorm or pre-training).

However, in some cases, we find that the performance drops after critic pre-training, although it is usually quickly recovered. Fundamentally, this arises because at the end of pre-training we change the objective of the actor from imitation learning (trying to choose the action that would have been chosen in the dataset), to off-policy reinforcement learnig (trying to choose the action that will maximise the return i.e. the critic prediction). If the values predicted by the critic are sufficiently accurate for the behaviour policy as a result of our proposed critic pre-training, then the performance should smoothly improve as we see in the Hopper environment. However, if the values are incorrect then the performance will drop. This is particularly the case for the HalfCheetah environment, and for EDAC on Walker2d. We hypothesise this is because all trajectories in the HalfCheetah-medium dataset end with a timeout (rather than termination) which is difficult to predict from the state alone in the current formulation of these environments (Pardo et al., 2022), making the Monte-Carlo values higher variance. We find that this can be mitigated by adding a small amount of TD target using

Equation 3, with $\lambda = 0.1$ for HalfCheetah, and Walker2d for EDAC (and 0 for all other cases). However, for more Markov environments, and for effectively episodic environments such as Hopper (since all but one of the medium-level trajectories end in termination) the addition of non-zero $\lambda$ can reduce the benefit of value error pre-training as shown in Appendix A.4.

As an ablation, we also compared with pre-training the actor with behaviour cloning alone, as shown in Appendix A.3. We found that while this provides good initial performance, performance quickly deteriorates after pre-training due to the inconsistent random critic, verifying our hypothesis regarding the performance drop and also the findings of (Orsini et al., 2021). We find pre-training durations of around 10-50k updates are generally sufficient for convergence, but optimal durations for each environment can be determined by monitoring the smooth convex supervised loss functions.

**Surprisingly, we notice that in many cases the final performance is also more stable**, even hundreds of thousands of updates after pre-training. As analysed in Fujimoto et al. (2022) and Chen & Jiang (2019), for finite data regimes such as the offline setting, the Bellman equation can be satisfied by infinitely many suboptimal solutions. Additionally, $Q$-values trained by minimising the Bellman error are often empirically found be to be inaccurate (Schulman et al., 2018). We hypothesise that this additional stability could be occurring because the initial pre-training using the value error reduces the subset of possible solutions to those with lower value error when subsequently minimising the Bellman error on the finite offline dataset. However, since this benefit is auxiliary to our central focus of improving efficiency, we leave a more theoretical investigation of this effect to further work.

## 5 EXTENSION TO DATA-LIMITED ADROIT ENVIRONMENTS

We now consider the Adriot tasks (Rajeswaran et al., 2018). These are more complex and realistic environments that require controlling a 24-DoF robotic hand to perform tasks such as aligning a pen, hammering a nail, opening a door, or relocating a ball. The human datasets provided for these environments are very limited, consisting of just 25 trajectories of human demonstrations. The cloned datasets augment these trajectories with a behaviour cloned policy to get a 50-50 mixture. These environments are suited to pre-training, since a successful policy is very difficult to learn from random, but the provided demonstrations can be improved by becoming more efficient. We also consider an AntMaze environment (Fu et al., 2021), where an ant robot must navigate between two points in a u-shaped maze, similarly requiring the policy to first learn to successfully navigate between points, and then improve the efficiency of motion without going too far out-of-distribution.

### 5.1 MOTIVATION FOR ACTOR AND CRITIC REGULARISATION WITH PRE-TRAINING

In such data-limited domains, off-policy algorithms often perform poorly and suffer from policy collapse as discussed in Section 3.1, since the actor or critic erroneously extrapolate out-of-distribution (OOD) of the offline data. While our approach brings both actor and critic in-distribution during pre-training, when we initially applied our pre-training approach from Section 4 to both TD3+BC and EDAC on the Adroit environments, we found that the pre-training performance (corresponding to behaviour cloning) often rapidly collapsed after pre-training, even with substantial regularisation. For performance improvement, critic values must be sufficiently accurate for policy actions to prevent chain extrapolation and performance collapse, requiring both to remain close to the behaviour distributions. However, most current offline RL methods only apply regularisation to only one of either the actor or the critic. Therefore we introduce two new algorithms that incorporate regularisation on *both* the actor and the critic to smoothly improve performance after pre-training in data-limited domains. First, we introduce TD3+BC+CQL, which combines the existing behaviour cloning on the actor with additional CQL-style regularisation on the critic to penalise large OOD $Q$-values. Second, we introduce EDAC+BC, which combines the existing uncertainty-based regularisation on the critic with additional behaviour cloning on the actor to penalise OOD actions.

### 5.2 IMPLEMENTATION DETAILS

As baselines, we consider both TD3+BC and EDAC without pre-training or additional regularisation, but with the same regularisation as TD3+BC+CQL and EDAC+BC for the existing regularisation components. We also consider the strong naive baseline of behaviour cloning, and CQL. For all baselines we use their benchmarked CORL implementations (Tarasov et al., 2022) and previously

published hyperparameters where possible. Full details of our hyperparameters are provided in Appendix A.6. We train all algorithms for 500k updates (corresponding to around 6 hours training time on average on our RTX2080 GPUs). For our novel algorithms, we pre-train for 200k steps, to provide sufficient time for both supervised convergence and offline RL improvement. Since performances are generally noisy and can oscillate between behaviour cloned performance and random performance due to policy collapse, in order to fairly measure performance within 500k updates we average online performance evaluated every 1000 offline updates between 200k and 500k updates. We evaluate all approaches using this procedure for 4 independent runs to estimate the variance.

## 5.3 RESULTS AND ANALYSIS

| Env-Dataset | BC | CQL | TD3+BC | EDAC | Pre-Trained TD3+BC+CQL (Ours) | Pre-Trained EDAC+BC (Ours) |
|---|---|---|---|---|---|---|
| pen-human | $61.9 \pm 8.3$ | $47.7 \pm 10.1$ | $51.1 \pm 13.6$ | $1.9 \pm 2.8$ | $\mathbf{70.4 \pm 8.5}$ | $\mathbf{73.7 \pm 13.6}$ |
| door-human | $0.6 \pm 0.4$ | $\mathbf{1.6 \pm 0.9}$ | $0.0 \pm 0.0$ | $-0.1 \pm 0.0$ | $0.3 \pm 0.2$ | $\mathbf{1.7 \pm 0.7}$ |
| hammer-human | $\mathbf{1.6 \pm 0.5}$ | $1.2 \pm 0.4$ | $0.8 \pm 0.3$ | $0.1 \pm 0.1$ | $\mathbf{1.5 \pm 0.6}$ | $\mathbf{1.5 \pm 0.4}$ |
| relocate-human | $0.1 \pm 0.0$ | $0.1 \pm 0.0$ | $0.0 \pm 0.0$ | $0.0 \pm 0.0$ | $0.1 \pm 0.1$ | $\mathbf{0.2 \pm 0.1}$ |
| pen-cloned | $49.0 \pm 10.4$ | $39.5 \pm 6.8$ | $16.0 \pm 4.9$ | $7.6 \pm 9.4$ | $\mathbf{59.0 \pm 8.3}$ | $\mathbf{57.2 \pm 10.3}$ |
| door-cloned | $-0.1 \pm 0.0$ | $\mathbf{0.7 \pm 0.4}$ | $-0.1 \pm 0.0$ | $0.0 \pm 0.0$ | $-0.1 \pm 0.0$ | $0.1 \pm 0.2$ |
| hammer-cloned | $0.6 \pm 0.3$ | $\mathbf{0.8 \pm 0.3}$ | $0.4 \pm 0.2$ | $0.2 \pm 0.0$ | $0.5 \pm 0.2$ | $\mathbf{0.7 \pm 0.3}$ |
| relocate-cloned | $0.0 \pm 0.0$ | $0.0 \pm 0.0$ | $-0.2 \pm 0.0$ | $-0.1 \pm 0.0$ | $-0.1 \pm 0.0$ | $-0.1 \pm 0.0$ |
| antmaze-umaze | $50.4 \pm 3.7$ | $51.9 \pm 3.7$ | $49.8 \pm 4.1$ | $0.0 \pm 0.0$ | $\mathbf{62.0 \pm 7.2}$ | $52.4 \pm 8.1$ |

Table 2: Normalized average returns on D4RL Adroit tasks, calculated by averaging performance between 200k and 500k updates for 4 independent runs. Our combined algorithms ensure that *both* the actor and critic are regularised to stay close to the behaviour policy after pre-training, and lead to greater performance than the component algorithms when learning from limited demonstrations.

We find that behaviour cloning (BC) provides a strong baseline, notably greatly outperforming previously quoted BC performances on these environments due to our addition of LayerNorm. We see that CQL and EDAC (with LayerNorm) perform reasonably with this evaluation approach using the same hyperparameters as specified in the original papers (Kumar et al., 2020; An et al., 2021), although could benefit from a greater training budget. The original TD3+BC paper did not consider the Adroit environments, but we see comparable performances with relatively strong behaviour cloning regularisation (tuned with $\alpha = 1$ for the pen environments and $\alpha = 0.1$ for the other environments). However, **our additions of CQL regularisation to TD3+BC and BC regularisation to EDAC both generally lead to improved performance across environments**. We find similar performance in benefits in the AntMaze-umaze domain, which is encouraging given the very different challenges provided by this environment. On ablating the pre-training stage, we found that the performance benefits observed were due to the additional regularisation components rather than pre-training, but the pre-training stage was beneficial for efficiency and identifying regularisation hyperparameter values that did not collapse performance after pre-training. Performance plots in these domains for the human and umaze datasets are provided in Appendix A.7.

## 6 CONCLUSION

We have demonstrated that appropriately pre-training policies and value-functions to first be consistent with the provided offline dataset can improve the efficiency and stability of subsequent off-policy reinforcement learning. In particular, first pre-training a critic or value network to minimise the Monte-Carlo value error removes much of the inefficiency and instability associated with subsequent bootstrapped temporal difference losses, and can more than halve the number of updates and associated computation required for state-of-the-art offline reinforcement learning algorithms to converge on standard environments. We also demonstrated that by combining appropriate regularisation on both policy and value networks can enable greater performance than policy or value regularisation alone at learning from limited human demonstrations on the challenging Adroit tasks. We hope that this research inspires further work on improving the efficiency and stability of offline reinforcement learning algorithms as the scale of offline learning continues to increase.

ETHICS STATEMENT

Our work considers making offline reinforcement learning more computationally efficient. This could save significant cost and emissions due to un-initialised bootstrapping for training large offline reinforcement models, which have recently been increasing in scale. The authors are not aware of any additional ethical concerns due to our work over existing methods.

REPRODUCIBILITY STATEMENT

Our codebase, including configuration files for all experiments contained within the paper, will be open-sourced if accepted.

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

# A  APPENDIX

## A.1  RATIONAL FOR SEPARATION OF ACTOR AND CRITIC PRE-TRAINING FOR ENTROPY REGULARISED REINFORCEMENT LEARNING

In section 3.3 and algorithm 2 of the main text, we propose separating the pre-training of the policy and value network into two separate phases for entropy-regularised reinforcement learning algorithms. By first pre-training the policy with soft behaviour cloning, an approximate behaviour policy can be learned which then enables approximate behaviour entropy bonuses to be included in the subsequent pre-training of the critic. However, an alternative approach could be to pre-train the policy and critic in parallel, as in the pure return maximisation framework. In theory this would require updating the returns-to-go for each policy update to incorporate the changing entropy bonuses as the variance of the policy is updated. Since this requires a complete forward pass of the policy for all subsequent states in the trajectories of the states sampled for an update, this makes the pre-training infeasibly expensive.

Another potential approach is to only train the mean of the standard Gaussian policy to match the behaviour policy, and keep the variance constant such that all entropy bonuses could be caluclated an intialisation and would be unaffected by policy pre-training. However, we note that the stadard tanh squashing applied to the policy to keep the sampled action within the environment action bounds leads to a changing entropy of the resulting policy, even with the Gaussian variance kept constant.

Another approach could be to separate the policy and value-pretraining, but continue the policy behaviour cloning during value-pretraining. While this is possible, for the entropy bonuses to be sufficiently accurate, the policy behaviour cloning must have effectively converged before value pre-training, meaning continuing to pre-train the policy in parallel with the critic has little effect and requires unecessary computation time.

A final approach we considered was to compute the soft returns-to-go based on the initialisation policy, and then only pre-train the values (no behaviour cloning). While this approach was successful and led to training efficiency gains, the rapid updating of the actor at the beginning of training (and particularly the rapidly changing policy entropy) quickly leads to inconsistent values, so we found that the investment in pre-training the policy with soft behaviour cloning first was worth the computational time in most cases.

## A.2 INVESTIGATION INTO AFFECT OF LAYERNORM

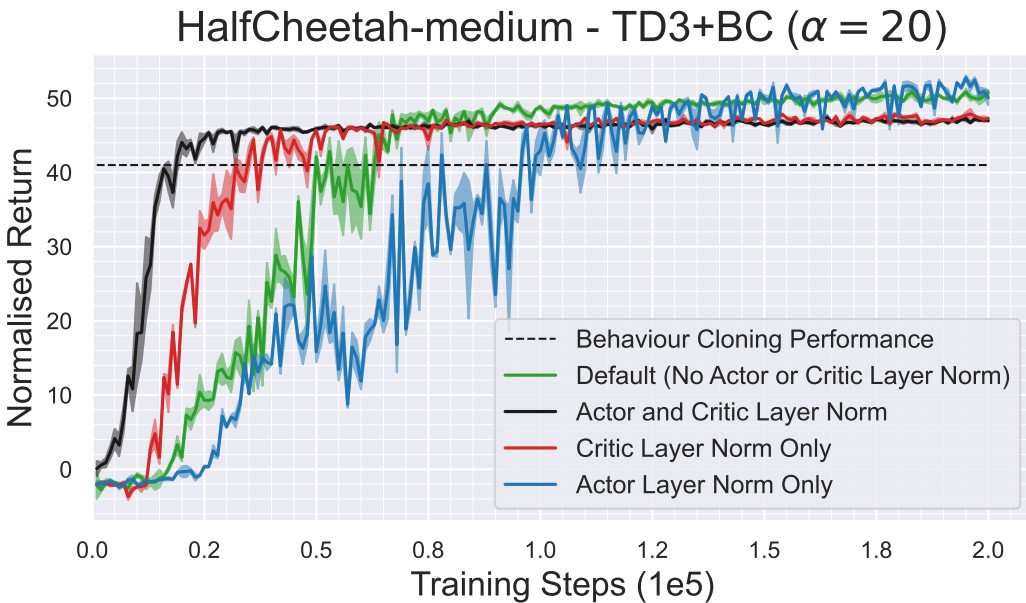

Figure 3: Investigation into the affect of adding LayerNorm to both the actor and critic networks for TD3+BC on HalfCheetah-medium.

We investigate the effect of the addition of LayerNorm (Ba et al., 2016) to both the actor and critic networks for TD3+BC on the HalfCheetah-medium dataset. The standard author implementation of TD3+BC (along with that of SAC-N, EDAC and most other off-policy reinforcement learning algorithms (Tarasov et al., 2022)) does not include any form of representational normalisation, and is shown in green. We consider the addition of LayerNorm after every linear linear layer in the network (before activation) except the final linear layer. We find that adding LayerNorms to the critic network leads to significant improvement in training efficiency and stability. This independently verifies the findings of Ball et al. (2023), who hypothesise that this occurs because the normalisation prevents severe value extrapolation for out-of-distribution actions, leading to overestimation error.

Surprisingly, we find that the addition of LayerNorm to the actor (without addition to the critic) leads to worse efficiency and stability than the default (no LayerNorm). This could be because normalising the state representation for the actor without doing the same for the critic leads to a form of state aliasing that prevents fine-grained action selection. However, the addition of LayerNorm to both the actor *and* the critic leads to greater training efficiency and stability than the default or applying either normalisation alone. We find that these insights generally hold across investigated environments and datasets. Therefore we decide to apply LayerNorm to both the actor and the critic for all experiments (except where explicitly stated otherwise) in our mission to improve training efficiency.

However, we notice this this addition comes at the cost of a few percent in final performance. While not shown here, we found that while the improvement in efficiency and stability due to the introduction of LayerNorm was generally consistent across D4RL MuJoCo environments and datasets, this cost to performance only occurred for the HalfCheetah environment. Therefore we expect the addition of LayerNorm to be universally introduced going forwards, and removed in the infrequent cases where it is found to hurt performance.

## A.3 ABLATION OF CRITIC PRE-TRAINING

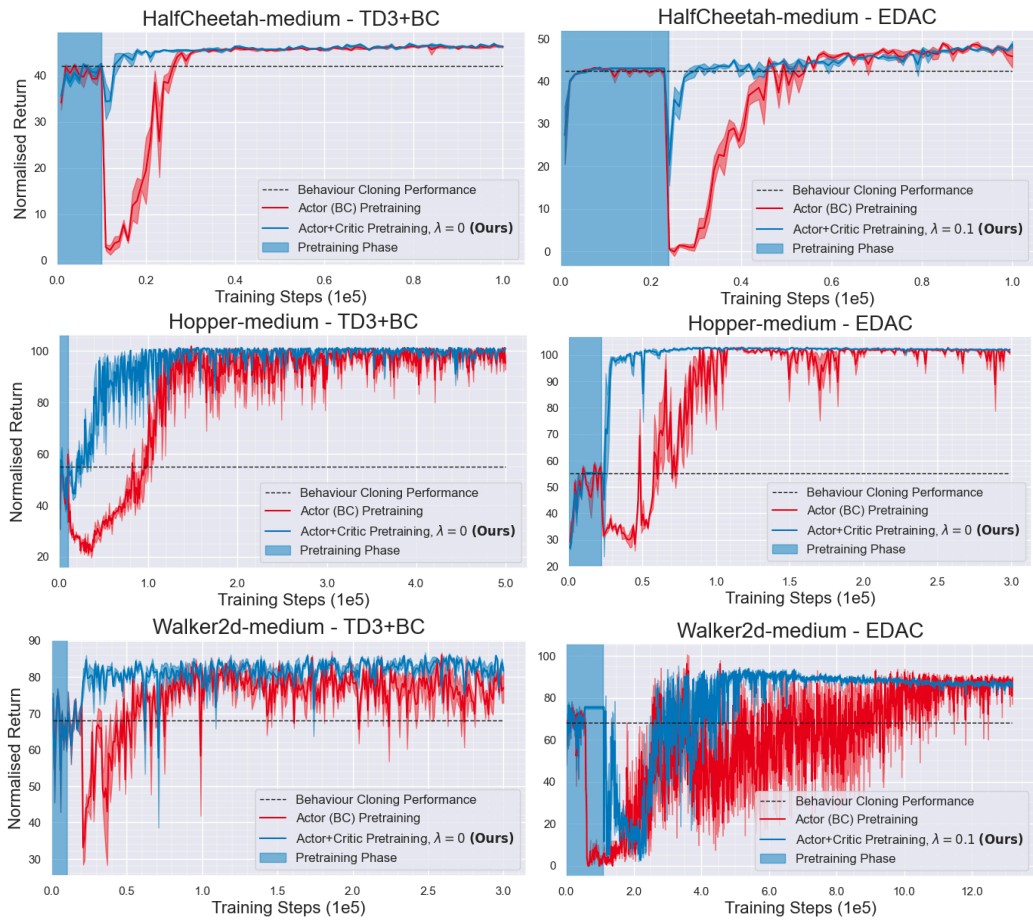

Figure 4: Ablation of critic pre-training to investigate whether the improved performance is due to our proposed critic pre-training.

We see that when pre-training the actor with pre-training, the initial performance matched our proposed actor and critic pre-training, but quickly declines after pre-training due to the randomly initialised critic, matching the findings of Orsini et al. (2021). Both implementations utilise LayerNorm (Ba et al., 2016). Therefore we see that the improved efficiency and stability from pre-training arises due to our combined actor and critic pre-training.

A.4  INVESTIGATION INTO EMPIRICAL BIAS VARIANCE TRADEOFF

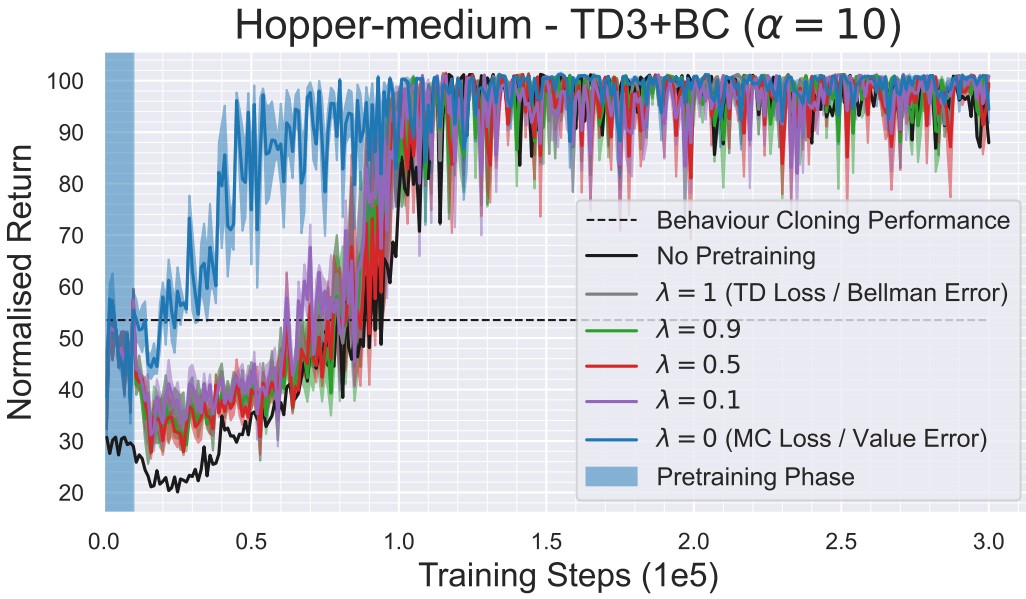

Figure 5: Investigation into the empirical bias-variance tradeoff by varying $\lambda$ defined in equation 3 for TD3+BC on Hopper-medium.

We investigate the bias-variance tradeoff described in section 3.2 in practice by empirically varying $\lambda$ for TD3+BC on the Hopper-medium dataset, selected due to the significant performance improvement and additional stability found as a result of pre-training. We find that while all values of $\lambda \in [0, 1]$ provide a small efficiency benefit over no pre-training, for all but $\lambda = 0$ (corresponding to the originally proposed value-error pre-training) this benefit is small, likely because even for $\lambda = 0.1$ the temporal difference component to the loss can have significant impact on the training dynamics, and the pre-training duration is not long enough for this bootstrapping loss to reach consistency. However for $\lambda = 0$ we observe the significant pre-training benefits described in the paper.

We also see that the greater stability on convergence only occurs for $\lambda = 0$, supporting our hypothesis in section 4 that this additional stability follows from the use of the value error (rather than the temporal difference error) in pre-training.

## A.5 MuJoCo Medium-Replay Dataset Experiments

We apply our pre-training approach proposed in Section 3 with an identical experimental implementation to that described in Section 4.1.

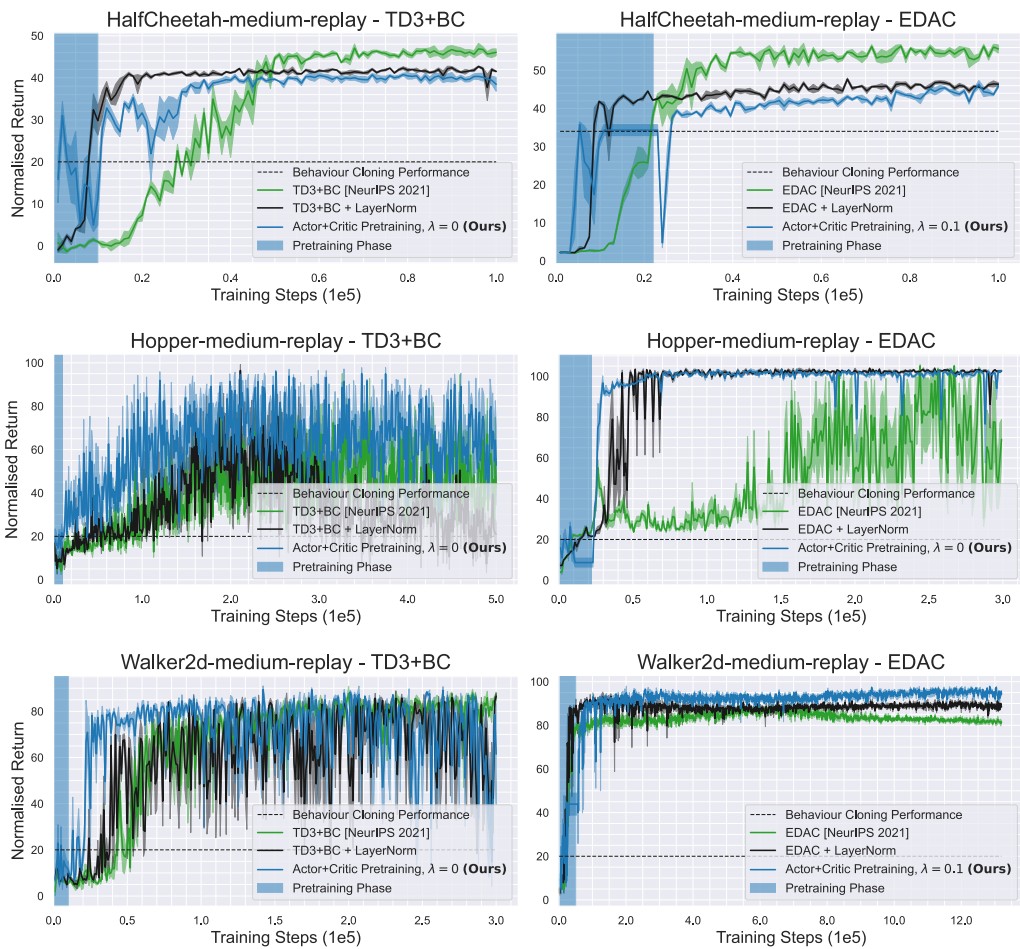

Figure 6: Application of supervised pre-training to the *medium-replay* MuJoCo datasets. We generally see training efficiency gains similar to those observed in Figure 6. Plots show mean and standard deviation at each timestep for 3 independent runs.

While we generally see similar efficiency gains to those observed for the medium datasets in Figure 6, we see there are no efficiency gains to be made on the HalfCheetah environments (given the baselines with our addition of LayerNorm converge so quickly) and that the performance of Walker2d for EDAC is much cleaner, likely due to greater diversity of data helping to stabilise performance.

A.6 KEY HYPERPARAMETERS FOR ADROIT EXPERIMENTS

The Adroit experiments were performed and evaluated as described in Section 5 of the main paper. Key hyperparameters for each algorithm are provided in Table 1 below. Full hyperparameter configurations (including those not provided below) are available in the config files of the provided CORL codebase (Tarasov et al., 2022). Where possible, hyperparameters were chosen to match previously published values for the Adroit environments, and otherwise their default implementation values. Crucially, where hyperparameters are shared between algorithms (such as between TD3+BC and TD3+BC+CQL) they were chosen to be equal, to investigate whether additional regularisation on the critic/actor can improve over the exisiting tuned regularisation on the actor/critic alone.

To incorporate our additional regularisation losses which may be of different scales to the existing losses, we utilise the normalisation strategy described in TD3+BC (Fujimoto & Gu, 2021). Namely for primary loss $\alpha$ and additional auxiliary loss $\beta$ which may be of a different scale, we combine them as follows to more evenly balance the losses throughout training:

$$\mathcal{L} = \alpha/|\alpha| + c\,\beta/|\beta| \tag{7}$$

where $|\cdot|$ denotes the magnitude of the gradient-detached loss and $c$ is the regularisation coefficient referred to as CQL/BC-regularizer provided in Table 1.

Finally, we note that for the behaviour cloning baseline and for the behaviour cloning regularisation in both TD3+BC(+CQL) and EDAC+BC we use 'hard' behaviour cloning (using a mean-squared error objective). In the case of the BC baseline and EDAC+BC it would be possible to use 'soft' behaviour cloning as in Equation 6, but we found in both cases 'hard' behaviour cloning (using the sampled action from the Gaussian policy for EDAC) performed much better. However we still use 'soft' behaviour cloning for pre-training EDAC+BC to maintain the policy entropy in pre-training.

Table 3: Adroit Experiments Key Hyperparameters

| ALGORITHM | TASK | PARAMETER | VALUE |
|---|---|---|---|
| BC | All | BC Objective | MSE |
| CQL | All | n-actions | 10 |
| CQL | All | Temperature | 1.0 |
| TD3+BC | Pen | $\alpha$ | 1.0 |
| TD3+BC | Door/Hammer/Relocate | $\alpha$ | 0.1 |
| TD3+BC | AntMaze | $\alpha$ | 5.0 |
| EDAC | Pen | N (num critics) | 20 |
| EDAC | Pen (Human) | $\eta$ | 1000 |
| EDAC | Pen (Cloned) | $\eta$ | 10 |
| EDAC | Door/Hammer/Relocate | N (num critics) | 50 |
| EDAC | Door/Hammer/Relocate | $\eta$ | 200 |
| EDAC | AntMaze | N (num critics) | 50 |
| EDAC | AntMaze | $\eta$ | 10 |
| TD3+BC+CQL | Pen | $\alpha$ | 1.0 |
| TD3+BC+CQL | Pen | CQL-regularizer | 1.0 |
| TD3+BC+CQL | Door/Hammer/Relocate | $\alpha$ | 0.1 |
| TD3+BC+CQL | Door/Hammer/Relocate | CQL-regularizer | 10.0 |
| TD3+BC+CQL | AntMaze | $\alpha$ | 5.0 |
| TD3+BC+CQL | AntMaze | CQL-regularizer | 1.0 |
| TD3+BC+CQL | All | n-actions | 10 |
| TD3+BC+CQL | All | Temperature | 1.0 |
| EDAC+BC | Pen | N (num critics) | 20 |
| EDAC+BC | Pen (Human) | $\eta$ | 1000 |
| EDAC+BC | Pen (Cloned) | $\eta$ | 10 |
| EDAC+BC | Door/Hammer/Relocate | N (num critics) | 50 |
| EDAC+BC | Door/Hammer/Relocate | $\eta$ | 200 |
| EDAC+BC | AntMaze | N (num critics) | 50 |
| EDAC+BC | AntMaze | $\eta$ | 10 |
| EDAC+BC | All | BC Objective | MSE |
| EDAC+BC | All | BC-regularizer | 1.0 |

## A.7   PERFORMANCE PLOTS FOR ADROIT EXPERIMENTS

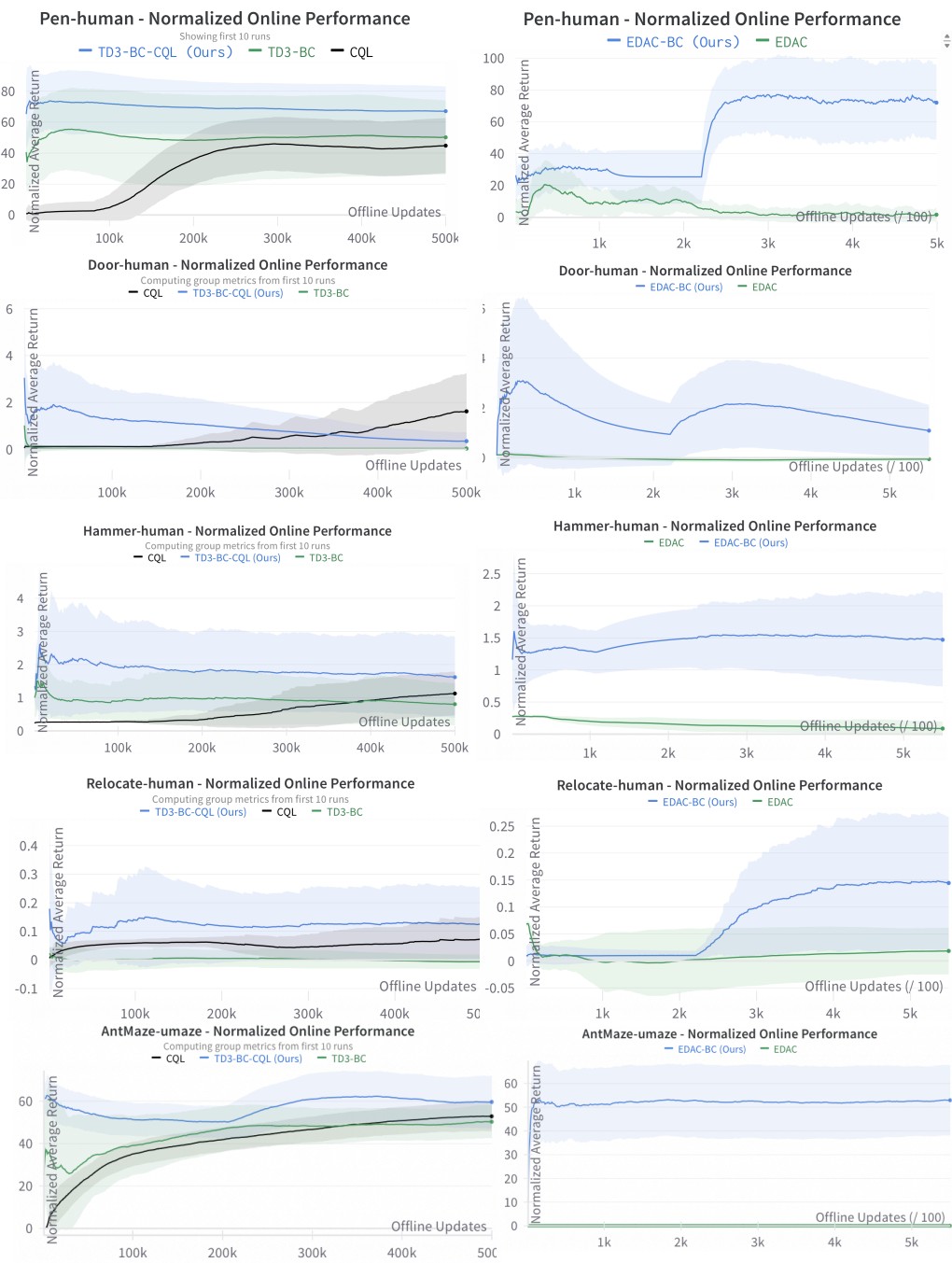

Figure 7: Training plots for Adroit environments using the human datasets, and the Antmaze-umaze dataset. To prevent performance collapse after pre-training we add additional regularisation. We find that combining actor and critic regularisation leads to better performance than equivalent actor or critic regularisation alone (regularisation hyperparameters provided in Table 3). However, we see that in many environments, subsequent off-policy reinforcement learning is not able to improve upon the initial pre-training performance corresponding to imitation learning (with LayerNorm). Plots show mean and standard deviation at each timestep for 4 independent runs.

## A.8 Outlook and Discussion

For academics and RL practitioners with a modest computational budget, the application of the proposed pre-training approach could significantly speed up research and development time, enabling more ideas to be investigated (Togelius & Yannakakis, 2023). For larger computational budgets and datasets, the proposed pre-training approach could save many thousands of GPU hours spent on un-initialised bootstrapping with associated cost and emissions, which is becoming of increasing importance for training large models (Patterson et al., 2021; Wu et al., 2022; Luccioni et al., 2022). Given that offline reinforcement learning currently appears to be the most promising avenue to scaling reinforcement learning and achieving associated emergent properties witnessed in related domains (Kumar et al., 2022a; Reed et al., 2022; Agarwal et al., 2020), we anticipate the scale of offline reinforcement learning to only increase.

