# OpenReview forum: "Efficient Offline Reinforcement Learning: The Critic is Critical"
_ICLR.cc/2024/Conference — Submitted to ICLR 2024_

### Official Review · Reviewer_p7js · 2023-10-24

**Soundness:** 3 good
**Presentation:** 2 fair
**Contribution:** 2 fair
**Rating:** 5
**Confidence:** 3

**Summary:**

The paper presents an approach that combines supervised learning and off-policy reinforcement learning to enhance efficiency and stability. This is achieved through pre-training the critic using a supervised Monte-Carlo value-error and applying regularization to both the actor and the critic. The results demonstrate a reduction in training time and improved learning efficiency.

**Strengths:**

* Utilizing Monte-Carlo estimation as the initialization for offline RL is reasonable, yet it is ignored in prior works.
* The efficacy of the proposed method is demonstrated through experiments conducted on MuJoCo and Adroit tasks.
* Implementation details are provided in the Appendix.

**Weaknesses:**

* First of all, the overall structure and writing of this paper necessitate meticulous reorganization and refinement. Some paragraph is confusing and hard to follow due to poor organization. Especially in Section 4 and Section 5, the important conclusion in these paragraphs need to be highlighted and summarized. In Section 5, the transition from Monte Carlo (MC) pretraining to emphasizing both actor and critic regularization is perplexing, especially since these regularization are not introduced in the methods section. And the title "Application to Adroit Environments" is incongruous as the methodology differs from the prior parts.

* The two parts of pretraining and regularization that the authors want to underscore appear to be incremental additions rather than naturally integrated components. This disjointed presentation detracts from the coherence of the paper and needs to be addressed.

* Regarding the methodology, the paper's primary emphasis appears to be on the use of Monte Carlo (MC) estimates as pretraining targets. However, the results in Appendix A.3 indicate that the efficiency and performance during pretraining stem from the Behavioral Cloning (BC) loss rather than MC. MC only contributes to stability during subsequent fine-tuning. Furthermore, in Section 5, the authors assert that pretraining is less critical than both regularization techniques. Consequently, I am unconvinced about the significance of this work.

*  The experiments are also limited in variety of dataset types and domains. For instance, BC pretraining may depend on data quality; therefore, additional dataset types such as "medium-replay" and "random" datasets are necessary to substantiate the importance of this work. Moreover, further ablation studies are required to validate the paper's claims. Current results in Figures 3, 5, and 6, which only base on a single dataset, are unconvincing. Additionally, the inclusion of more domains, such as AntMaze, would be beneficial.

**Questions:**

* Revise the structure and refine the writing to make the conclusions and emphasized information more evident.

*  The two parts of the methods seem to be added incrementally rather than integrated as natural components. The authors need to improve the presentation to address this problem.

* The reviewer finds the significance of the MC pretraining unconvincing based on the results in Appendix A.3 and Section 5.

* How is the performance of the pretraining approach on "medium-replay" and "random" datasets?

* It is necessary to conduct more ablation experiments with additional environments.

* How does pretraining and the regularizations perform on the AntMaze domain?

---

> ### Author Response · Authors · 2023-11-22
> **Author Response to Reviewer p7js (Part 1/2)**
>
> We thank you for your honest review and helpful feedback, which we have now incorporated to improve the coherence and completeness of our paper. In particular, we have attempted to address each of your concerns/weaknesses as follows:
>
> > Revise the structure and refine the writing to make the conclusions and emphasized information more evident.
>
> We have restructured and re-written Sections 4 and 5 following your advice to better separate our experimental setup and implementation details from the results and discussion in each section. Each paragraph now corresponds to a separate point of analysis, and we have highlighted the key conclusions in each section. We have also renamed Section 5 from ‘Application to Adroit Environments’ to ‘Extension to Data-Limited Adroit Environments’, and added a Section ‘Motivation for Actor and Critic Regularisation with Pre-Training’ to explain the necessary extension of our methodology to handle the limited-data setting considered in this section.
>
> > The two parts of the methods seem to be added incrementally rather than integrated as natural components. The authors need to improve the presentation to address this problem.
>
> As mentioned above, we have now added an additional Section (5.1) to better justify our addition of combined actor and critic regularisation with pre-training. To summarise here, pre-training trains the actor and critic networks to first match the behaviour policy distributions, which improves the efficiency of subsequent off-policy RL, as we show in Section 4. However, in data-limited domains, subsequent off-policy RL can lead to policy collapse, where the performance drops to close to zero after pre-training. To prevent this, and to try to maintain at least the pre-training performance, we can add additional regularisation to both the actor and the critic to prevent chain extrapolation away from the initialised behaviour distributions. Therefore, our proposal is to pre-train both the actor and critic to match the behaviour policy, and then regularise them to prevent extrapolation significantly out-of-distribution of the behaviour distributions.
>
> While we provide this additional justification of additional regularisation in Section 5.1, we also anticipate and highlight this idea throughout the paper, with the addition of regularisation being mentioned in the last paragraph of the introduction (Section 1), the overview of our approach in Section 3 (including Algorithms 1 and 2), and the abstract and conclusion.
>
> > The reviewer finds the significance of the MC pretraining unconvincing based on the results in Appendix A.3 and Section 5.
>
> Appendix A.3 provides an ablation which isolates the significance of the MC pre-training, so we are confused about why the reviewer is unconvinced of the significance based on these results.
>
> In Figure 2, we see that pre-training both the actor with imitation learning and the critic with a Monte-Carlo value error (blue) provides an efficiency benefit over randomly initialising these networks (black, still using additional LayerNorm). However, one might wonder if pre-training the actor is enough, since this is what leads to non-zero performance during pre-training (and does not depend on the critic values at all, since it is only trying to imitate and not optimise behaviour during this stage).
>
> Therefore, in Appendix A.3, we perform actor only pre-training with imitation learning. We see that during pre-training performance is equivalent as expected, but after pre-training performance drops off and takes a long time to recover, generally performing similarly to training from scratch with TD-learning as in Figure 2. Therefore, since the blue line reaches maximal performance more quickly than both the black line in Figure 2 and the red line in Figure 4 (Appendix A.3) we have isolated that the significant efficiency gain is due to our proposed MC pre-training.
>
> In Section 5, we agree that pre-training does not help in these environments (as we mention) since performance collapses with or without pre-training. Therefore it is the combined regularisation approach that we propose that is responsible for the performance gains over the baselines in these data-limited environments. However, the pre-training stage is still beneficial for obtaining a non-zero initial baseline performance that allows for easier tuning of the regularisation parameters to prevent subsequent performance collapse.

---

> > ### Author Response · Authors · 2023-11-22
> > **Author Response to Reviewer p7js (Part 2/2)**
> >
> > > How is the performance of the pretraining approach on “medium-replay” and “random” datasets?
> >
> > As we mention at the beginning of Section 4, we only expect our pre-training approach to provide benefits in the case of medium datasets (or more realistically, human datasets) where there’s a benefit to first imitating the behaviour policy, but still room for improvement. For random data, pre-training leads to an uninformed initialisation that does not provide any efficiency benefits, while for expert data, pre-training already provides optimal performance, so there is no benefit from subsequent reinforcement learning (both of which we verified empirically during our initial research).
> >
> > However, following your suggestion, we have now added results from the medium-replay datasets in Appendix A.5. We find similar performance improvements across environments as for the medium-datasets, demonstrating that it is still possible to learn useful information from pre-training on mixed policy datasets.
> >
> > > It is necessary to conduct more ablation experiments with additional environments.
> >
> > We appreciate that performing all ablations on all datasets and environments would be ideal, but we do not believe this is necessary given that we have already run our most important ablations across all considered environments.
> >
> > For example, while Figure 3 in Appendix A.2 only considers one environment to separate the effect of LayerNorm on the actor and the critic, we ablate the effect of LayerNorm vs no LayerNorm on all environments in Figure 2. We break this down further in the HalfCheetah environment in Appendix A.2 because this environment was the only one which showed evidence of LayerNorm limiting convergence performance. Similarly, we only show intermediate values of lambda for one environment in Figure 5, but we show $\lambda=0/0.1$ and $\lambda=1$ (after pre-training) for all environments in Figure 4 in Appendix A.3.
> >
> > Finally, since we now have complete results for the medium-replay datasets in Appendix A.5 following your feedback, we have removed the final ablation you mention (Figure 6) which previously highlighted the empirical promise of applying our approach to mixed-policy datasets.
> >
> > > How does pretraining and the regularizations perform on the AntMaze domain?
> >
> > Following your feedback, we have now added the AntMaze (umaze) domain as an additional experiment to Section 5, since this indeed provides another suitable environment/dataset where imitation provides a non-trivial performance but this can be improved upon with off-policy reinforcement learning. We use an identical training procedure to that used for the Adroit environments, and show similar performance benefits of our pre-trained and regularised approaches, with relevant hyperparameters added to Table 3 in Appendix A.5.
> >
> >
> > Thank you again for your helpful review and suggestions. We hope that our responses and additional experiments have addressed your concerns and substantiated the importance of our work towards understanding and improving the efficiency of offline reinforcement learning.

---

> > > ### Comment · Reviewer_p7js · 2023-11-23
> > > **Thanks for your response**
> > >
> > > Thank you for your response. I appreciate the inclusion of additional results and modifications, which have undoubtedly improved this work. As a result, I am inclined to increase my score to 5. The paper requires further meticulous refinement in its writing to better structure the methodology. Specifically, the regularization part seems to be specifically tailored for adroit environments, which makes its organization stand apart.

---

### Official Review · Reviewer_KvJB · 2023-10-31

**Soundness:** 3 good
**Presentation:** 2 fair
**Contribution:** 2 fair
**Rating:** 5
**Confidence:** 3

**Summary:**

Off-policy reinforcement learning is able to further improve the offline RL performance while suffering from instability and inefficiency. This works propose to bridge the supervised approach and the off-policy approach aiming for a more stable offline RL. The key innovation is the pre-training of the critic using a supervised Monte-Carlo value-error, which leverages information from offline data. This step provides a consistent actor and critic for off-policy TD-learning. The experiments on D4RL MuJoCo benchmark show that the proposed method is more stable and efficient during the offline training comparing with other method, such as behavior cloning and TD3.  Meanwhile the results in Adroit shows the proposed method can achieve good performance for most of the tasks.

**Strengths:**

1. This paper is well-motivated and focuses on an important problem in offline RL.
2. The proposed method is easy to understand and shown to perform well comparing with previous methods.

**Weaknesses:**

1.  The motivation example is rather unnecessary due to its simplicity
2.  The consistence of the actor and critic networks play critical roles in the proposed method, while it is unclear how much degree of consistence is needed in order to make it work well for off-policy training? If it is possible to derive any explicit criteria on this matter?

**Questions:**

1. How do you choose the pretraining phase steps, e.g. different environments choose different pretraining steps in Figure 2. What is the impact of the pretraining steps on the final performance?
2. What is the main reason for the huge performance drop in Walker2d-edium-EDAC?
3. What is the computation complexity for the pretraining phase? Does it exceed a lot comparing with the offline training?

---

> ### Author Response · Authors · 2023-11-22
> **Author Response to Reviewer KvJB (Part 1/2)**
>
> Thank you for your review of our work. We’re pleased that you recognise the importance of our motivation and the simplicity and effectiveness of our method for improving the efficiency of offline RL.
>
> To address your questions first:
>
> > How do you choose the pretraining phase steps, e.g. different environments choose different pretraining steps in Figure 2. What is the impact of the pretraining steps on the final performance?
>
> The number of pre-training steps can be considered a hyperparameter which we set to between 10 and 50 epochs of the data depending on the environment in Figure 2. However, since both the actor and critic objectives in this pre-training phase are supervised objectives fit with a mean squared error, the loss is smooth and convex, so it is straightforward to see when these losses have converged to move onto the subsequent offline reinforcement learning objective. If the number of pre-training steps is fewer than required for convergence, this reduces the subsequent training efficiency to somewhere between the red (no critic pre-training) and the blue (pre-training steps required for convergence) performance lines in Figure 4 in Appendix A.3 (i.e. the performance would drop after pre-training). If the number of pre-training steps is greater than required for convergence, this just leads to unnecessary updates (effectively extending the width of the blue region with no benefit).
>
> > What is the main reason for the huge performance drop in Walker2d-edium-EDAC?
>
> Fundamentally, this arises because at the end of pre-training we change the objective of the actor from trying to choose the action that would have been chosen in the dataset (imitation learning), to trying to choose the action that will maximise the return i.e. critic prediction (off-policy reinforcement learning). If the values predicted by the critic are sufficiently accurate for the behaviour policy as a result of our proposed critic pre-training, then the performance should smoothly improve as we see in the Hopper environment. However, if the values are incorrect then the performance will drop, as we see in the extreme case for the red lines in Figure 4 in Appendix A.3 where the critic is not pre-trained at all.
>
> In the Walker2d environment, we have an intermediate case where the values have been pre-trained, but are high enough variance that sometimes an action is selected which is erroneously expected to have a high value but it does not, leading to a performance drop before the critic is updated to correct its value prediction. This is an example of the bias-variance tradeoff, as we discuss in Section 3.2. The values are generally higher variance for EDAC than for TD3+BC because the critic must predict soft Q values, which also incorporate an entropy bonus as defined by Equation 5. We believe the HalfCheetah and Walker environments are higher variance because the medium trajectories end with timeouts rather than termination (as for all but one trajectories in Hopper-medium) which are more difficult to  predict without access to the current timestep [1], as we mention in Section 4. The Walker2d-medium-EDAC case therefore has the highest variance because it uses soft Q-values incorporating entropy, the episodes end with a difficult to predict timeout condition, but the episodes can still end in termination (leading to high variance on evaluation).
>
> To mitigate this performance drop, we can add in a temporal-difference component as in Equation 3, which reduces the variance at the cost of bias - in our case reducing the performance drop at the cost of efficiency. While we use a small temporal difference component for these environments ($\lambda=0.1$), for increasingly large datasets used for offline RL, we generally expect that the variance will be lower and a temporal difference component will not be required for pre-training.
>
> > What is the computation complexity for the pretraining phase? Does it exceed a lot comparing with the offline training?
>
> The pretraining phase has a lower computational complexity than the offline RL training (for the case that $\lambda=0$). This is because each pre-training update requires one forward and backward pass of the actor and critic networks to regress them towards their static supervised target. However, each offline RL training update requires an additional forward pass of the actor network for the subsequent timestep and then the target critic on the subsequent timestep in order to compute the temporal difference target ($Q(s_{t+1}, \pi(s_{t+1})$) for the critic to regress towards. This means each pre-training update is also generally faster than each offline RL update. If $\lambda \neq 0$, then the computational complexity is effectively equivalent.
>
> References:
>
> [1] Time Limits in Reinforcement Learning, https://arxiv.org/abs/1712.00378

---

> > ### Author Response · Authors · 2023-11-22
> > **Author Response to Reviewer KvJB (Part 2/2)**
> >
> > To address your concerns:
> >
> > > The motivation example is rather unnecessary due to its simplicity
> >
> > We believe that the motivational example is helpful to isolate the core idea and general applicability of our method before the implementation complexity of offline reinforcement learning algorithms and practicality of real environments and datasets are introduced in later sections. Reviewer NdGK also listed the motivational example as a strength of the paper to provide intuition for the method.
> >
> > > The consistence of the actor and critic networks play critical roles in the proposed method, while it is unclear how much degree of consistence is needed in order to make it work well for off-policy training? If it is possible to derive any explicit criteria on this matter?
> >
> > While it is difficult to derive equations that apply in practice for deep reinforcement learning algorithms, our empirical ablations attempt to address this matter. Figure 4 in Appendix A.3 demonstrates that when both the actor and the critic are initialised to the behaviour policy during pre-training as we propose, off-policy training follows the blue line. However, if the actor matches the behaviour policy (pre-trained with imitation learning), but the critic is randomly initialised (not at all consistent), then off-policy training follows the red line. Therefore these lines serve as empirical ‘bounds’ representing no consistency (random critic, red) and approximate consistency (pre-trained critic, blue), which other degrees of partial consistency should lie between.
> >
> > Thank you again for your helpful review. We hope we have addressed your questions and you can appreciate the value of our contribution towards understanding and improving the efficiency of offline reinforcement learning.

---

> ### Comment · Reviewer_KvJB · 2023-11-23
> **Thank authors for the response**
>
> I appreciate the authors effort on addressing my questions. Based on your response, my major concerns remain, in particular: 1) since this work is mainly focusing on the empirical study, I find many details are lacking (or not explained well). For instance, the details on the "consistence" of the A and C networks. Many further questions can be asked based on the paper, e.g., does inconsistent A C necessary will contribute to bad performance? 2) The pre-training step plays an important role on the learning performance and efficiency, which is treated as a hyper-parameter. It can be tricky for other people to utilize the results in practice. Overall I will keep my original evaluation on this work.

---

> > ### Author Response · Authors · 2023-11-23
> > **Thank Reviewer KvJB for the Response**
> >
> > Thank you for your response, and for acknowledging our effort. We'd just like to highlight that for 1) Inconsistency of the critic leads to poor performance as we show in Appendix A.3, where the performance of a pre-trained actor collapses because the critic values do not correspond to the behaviour policy, and 2) The pre-training steps should be sufficient for convergence to the behaviour policy, but as we mention can be determined from the critic loss so is straightforward to set compared to other hyper-parameters in offline reinforcement learning. However, we appreciate your feedback and thank you again for the review.

---

### Official Review · Reviewer_NdGK · 2023-11-02

**Soundness:** 3 good
**Presentation:** 3 good
**Contribution:** 2 fair
**Rating:** 5
**Confidence:** 4

**Summary:**

The authors in this paper consider pretraining the critic function using a mix of objectives of Monte-Carlo estimation and TD estimation of Q values from offline data and then train both the critic function and with policy with standard offline algorithms such as  TD3+BC. The empirical results demonstrate that such a training pipeline would help the latter training and make the policy learning converges faster.

**Strengths:**

- The authors show a toy example at the beginning of the paper, which demonstrates the intuition why the pertaining of the critic function might help for later offline policy and critic learning.
- The idea is simple to follow and not hard to implement.
- The authors evaluate the idea on both simple offline benchmark environments such as mujoco, and also hard ones, such as Adroit environments. Demonstrate that the proposed method can be helpful for both simple and complex scenarios.
- A detailed ablation study has been done in the appendix to make sure the proposed idea is valid and indeed helps the latter offline training.

**Weaknesses:**

- The pretraining stage increases the complexity of the overall training pipeline. From the training curve, we can see that the training converges faster than that without the pertaining, but almost for each environment, the performance would first drop and then begin to improve, which is quite weird in terms of robustness for the training.
- The authors did not provide the training curves of policy learning on hard environments, which makes me wonder if the performance drop would be even larger than that of standard mujoco environments.

**Questions:**

- Please explain the performance drop phenomenon in detail.
- Please provide the training curves on hard environments such as Adroit.
- Please explain why the pertaining steps are different for different environments, is that a hyperparameter?
- I wonder if the pertaining loss can be a regularization loss in additional to previous regularization loss, maybe in this way we can make sure the whole training is more robust and the training curve would be more smooth.

---

> ### Author Response · Authors · 2023-11-22
> **Author Response to Reviewer NdGK**
>
> We sincerely thank you for your thorough and helpful review. We’re pleased that you appreciated the simplicity and intuitiveness of the idea, the range of evaluation environments, and our ablations.
>
> To address your concerns and questions:
>
> > Please explain the performance drop phenomenon in detail.
>
> This is indeed an interesting phenomenon. Fundamentally, this arises because at the end of pre-training we change the objective of the actor from trying to choose the action that would have been chosen in the dataset (imitation learning), to trying to choose the action that will maximise the return i.e. critic prediction (off-policy reinforcement learning). If the values predicted by the critic are sufficiently accurate for the behaviour policy as a result of our proposed critic pre-training, then we expect the performance to smoothly improve as we see in the Hopper environment. However, if the values are inaccurate then the performance may drop, as we see in the extreme case for the red lines in Figure 4 in Appendix A.3 where the critic is not pre-trained at all.
>
> In the HalfCheetah and Walker environments, we have an intermediate case where the values have been pre-trained, but are high enough variance that sometimes an action is selected which is erroneously expected to have a high value but it does not, leading to a performance drop before the critic is updated to correct its value prediction. We believe the HalfCheetah and Walker (medium) environments are higher variance because the medium trajectories end with timeouts rather than termination (for all but one trajectory in Hopper-medium) which is more difficult to predict without access to the current timestep [1], as we mention in Section 4. This is an example of the bias-variance tradeoff, as we discuss in Section 3.2. To mitigate this performance drop, we can add in a temporal-difference component as in Equation 3, which reduces the variance at the cost of bias - in our case reducing the performance drop at the cost of efficiency. While we use a small temporal difference component for these environments ($\lambda=0.1$), for increasingly large datasets used for offline RL, we generally expect that the variance will be lower and a temporal difference component will not be required for pre-training.
>
> > Please provide the training curves on hard environments such as Adroit.
>
> We have now provided these training curves in Appendix A.7. There is no noticeable drop-off in performance in these environments. However, as we discuss at the beginning of Section 5, we initially found that the pre-training performance collapsed entirely on these environments due to the fact that the datasets are very limited, so the critic values were leading to actions that were very different to those found in the dataset and therefore collapsing performance as we discuss in Section 2.2. As a result of this initial performance collapse, we added additional regularisation to ensure both the actor and critic remained close to the behaviour policy and prevent the actor capitalising on the extrapolated critic variance. This prevents the actor from taking actions that are substantially different to the behaviour policy and therefore mitigates any performance drop after pre-training.
>
> > Please explain why the pertaining steps are different for different environments, is that a hyperparameter?
>
> Yes, the number of steps of pre-trainining can be considered a hyperparameter. However, since both the actor and critic objectives in this pre-training phase are supervised objectives fit with a mean squared error, the loss is smooth and convex, so it is straightforward to see when these losses have converged to move onto the subsequent offline reinforcement learning objective.
>
> > I wonder if the pertaining loss can be a regularization loss in additional to previous regularization loss, maybe in this way we can make sure the whole training is more robust and the training curve would be more smooth.
>
> Yes, we agree that this could be sensible and we tried using the Monte-Carlo values as regularisation ourselves at one point during our research. However, we found that the performance can still drop using these values, and the standard CQL loss regularisation was more effective at preventing performance drop due to large OOD Q values, so we applied this to TD3+BC to regularise the critic instead. This also makes the training curves more smooth, as now provided in Appendix A.7.
>
> Thank you again for your helpful review. We hope we have addressed your questions and you can appreciate the value of our contribution towards understanding and improving the efficiency of offline reinforcement learning.
>
> References:
>
> [1] Time Limits in Reinforcement Learning, https://arxiv.org/abs/1712.00378

---

### Author Response · Authors · 2023-11-22
**Author Summary of Improvements following Reviewer Feedback**

We thank all of the reviewers for taking the time to review our work and for their helpful feedback, which we have now incorporated into our paper. The main additions to our paper are as follows:

1. Restructured Sections 4 and 5 to improve the coherence of our paper and better motivate the need for our additional combined regularisation approaches in Section 5,
2. Added additional experiments for the medium-replay datasets in Appendix A.5,
3. Provided performance plots for our Adroit experiments in Appendix A.7,
4. Added additional experiments for Antmaze (umaze) in Section 5 using the same methodology as for the Adroit experiments.

We have also provided responses to each reviewer below to address individual questions and concerns.

---

### Meta-Review · Area_Chair_zrgj · 2023-12-08

**Metareview:**

The paper considers a Monte-Carlo pre-training of the critic in TD style methods for offline reinforcement learning, and find faster convergence and better final solutions (using classical methods) when this is implemented.
The method is simple and intuitive, and allows to solve offline RL problems `a bit better'.

Nonetheless, there are several concerns, such as an unexplained performance drop after pretraining, limited experiments, lack of theoretical grounds and exposition issues.

**Justification For Why Not Higher Score:**

The paper could be accepted

**Justification For Why Not Lower Score:**

N/A

---

### Decision · Program_Chairs · 2024-01-16

Reject